# Implementation of the ERAS (Enhanced Recovery After Surgery) protocol for hysterectomy in the Piedmont Region with an audit&feedback approach: Study protocol for a stepped wedge cluster randomized controlled trial. A study of the EASY-NET project

Elisa Piovano [1,2]*, Eva Pagano[3], Elena Del Piano[4], Federica Rinaldi[5], Valentina Palazzo[5], Paola Coata[6], Daria Bongiovanni[7], Monica Rolfo[8], Laura Ceretto Giannone[9], Deliana Veliaj[9], Marco Camanni[4], Andrea Puppo[10], Giovannino Ciccone[3], the ERAS-Gyneco Piemonte group[¶]

**1** Obstetrics and Gynecology Unit, Regina Montis Regalis Hospital–ASL CN1, Mondovì (Cuneo), Italy, **2** Obstetrics and Gynecology Unit n. 3, Sant'Anna Hospital, AOU Città della Salute e della Scienza di Torino, Torino, Italy, **3** Clinical Epidemiology Unit, AOU Città della Salute e della Scienza di Torino–CPO Piemonte, Torino, Italy, **4** Obstetrics and Gynecology Unit, Martini Hospital—ASL Città di Torino, Torino, Italy, **5** University of Torino, Torino, Italy, **6** Unit of Dietetic and Clinical Nutrition, Ordine Mauriziano Hospital, Torino, Italy, **7** Unit of Endocrinology, Andrology and Metabolism, Humanitas Gradenigo, Torino, Italy, **8** Healthcare Services Direction, Humanitas, Torino, Italy, **9** Anesthesiology and Intensive Care Unit, Humanitas, Torino, Italy, **10** Obstetrics and Gynecology Unit, Santa Croce e Carle Hospital, Cuneo, Italy

¶ Membership of the ERAS-Gyneco Piemonte group is provided in the Acknowledgments.
* piovano.elisa@gmail.com

## Abstract

### Introduction

ERAS (Enhanced Recovery After Surgery) is a perioperative program combining multiple evidence-based interventions designed to reduce the surgical stress response. Despite the publication of dedicated guidelines, ERAS application to gynecologic surgery outside clinical studies has been slow and fragmented. To promote the systematic adoption of the ERAS program in the entire regional hospital network in Piedmont an Audit-and-Feedback approach (A&F) has been adopted within a cluster randomized controlled trial, aiming to estimate the true impact of the protocol on a large, unselected population.

### Methods

The study protocol provides for a multicenter stepped wedge cluster randomized trial, focused on women undergoing an hysterectomy, for comparison between standard perioperative management and perioperative management according to the ERAS protocol. The primary outcome is the length of hospital stay (LOS). Secondary outcomes are: post-

**Data Availability Statement:** No datasets were generated or analysed during the current study. All relevant data from this study will be made available upon study completion.

**Funding:** This work was supported by the Italian Ministry of Health and the Piedmont Region as part of the Easy-Net Project, grant number NET-2016-02364191.

**Competing interests:** The authors have declared that no competing interests exist.

operative complications, quality-of-recovery at 24-hours after surgery, 30-day readmissions, patients' satisfaction, healthcare costs. The compliance to all the ERAS items is monitored with an A&F approach. All the gynecologic units of Piedmont hospitals are involved and all the patients hospitalized for elective hysterectomy in the period of the study are included. Centers, stratified by surgical volume and randomly assigned to four groups, are randomly ordered to activate the ERAS protocol in four periods, every three months. The planned calendar and the total duration of the study have been extended for six months due to the COVID-19 pandemic. The expected sample size of about 2400 patients has a high statistical power (99%) to detect a reduction of LOS of 1 day (effect size 0.5) and to estimate clinically meaningful changes in the other study endpoints. The study protocol has been approved by the Ethical Committee of all participating centers. Study results will be timely circulated within the hospital network and published in peer-reviewed journals.

## Conclusion

Results are expected to demonstrate positive clinical outcomes of the ERAS protocol even when its implementation is directed towards an entire regional network of gynecologic units, and not only towards selected and highly motivated centers.

## Trial registration

NCT04063072

## Introduction

ERAS (Enhanced Recovery After Surgery) is a perioperative program combining evidence-based pre-, intra- and post-operative interventions in order to reduce the surgical stress response and improve patient recovery [1].

In 2016 [2, 3] the ERAS Society developed guidelines for gynecology and gynecologic oncology surgeries that were updated in 2019 [4]. The principal items of the ERAS protocol for gynecologic surgery are reported as (S1 Table in S1 File).

Several recent studies reported that the application of the ERAS protocol to gynecologic surgery showed both clinical benefits (statistically significant reductions in hospital length of stay (LOS), opioid consumption, complications, 30-day readmissions and increased patient satisfaction) and cost savings to the healthcare system [5–8]. Moreover, some studies showed a positive dose-response relationship: the higher the adherence to the ERAS gynecology guidelines, the higher the clinical benefits [9, 10]. A recent systematic review of 31 studies (6703 gynecologic oncology patients) and meta-analysis demonstrated a decrease in LOS of 1.6 day, a 32% reduction in complications and a 20% reduction in readmission with ERAS implementation [11]. The mean cost savings for ERAS patients was $2129 USD [11].

However, validity of ERAS applied to gynecology has been questioned as available studies are based on sequential cohort designs (novel ERAS strategy against historical controls) and therefore prone to biases, including the Hawthorne effect (inadvertent introduction of an effect through the act of studying the intervention) [12, 13].

In a joint effort by the ERAS and ERAS USA societies, the Reporting on ERAS Compliance, Outcomes, and Elements Research (RECOvER) checklist was recently published [14] to provide a standardized framework for the designing and the reporting of ERAS-related studies.

A monocentric randomized study, compliant with the RECOvER checklist, was published in 2020 [15]. This study found that, compared with standard perioperative care, women in the ERAS group had a shorter LOS (median, 2 days versus 4 days), 15% less postoperative complications, and improved patient satisfaction at 24 hours and at discharge (based on the Quality of Recovery 15 questionnaire scores) [16]. The reduced LOS was confirmed in the prespecified analysis favoring the ERAS cohort in benign, malignant, open, and minimally invasive subgroups. However, the small sample size and the individual patient randomization within a single center may have limited the validity and generalizability of these findings.

In this evolving background, the application of the ERAS protocol in gynecologic surgery has been slow and limited to a minority of hospitals, at least in Italy [17] and Piedmont (a region of North West of Italy with 4.3 million population) (regional survey–unpublished data). In 2017 the diffusion of ERAS protocol in Piedmont was limited to few hospitals, particularly open to change, that had spontaneously adopted the protocol. This survey convinced the regional health authorities that, without a systematic implementation effort on the entire regional hospital network, the impact of ERAS on the overall quality on a regional scale was negligible and, possibly, a source of unjustified heterogeneity between centers and inequalities among patients.

To promote the systematic adoption of ERAS protocol in the entire regional hospital network in Piedmont an Audit and Feedback approach (A&F) was planned and a large, population based, cluster randomized controlled trial was designed to carefully estimate the effectiveness of the protocol on several quality endpoints.

The A&F strategy is recommended by the ERAS society guidelines for gynecologic/oncology surgery (quality of evidence: high; recommendation: strong) as an instrument to be applied regularly by healthcare providers when driving change or implementing ERAS programs [4].

This regional project is part of a larger project on the evaluation of the effectiveness of audit and feedback interventions (EASY-NET), a Network Project founded by the Ministry of Health and the participating Regions (NET-2016-02364191).

In this article we present the study protocol of the ERAS-Gyneco trial (http://www.clinicaltrials.gov, registration number: NCT04063072).

## Methods

### Trial design

The ERAS Protocol Implementation in Piedmont Region for Hysterectomy (ERAS-Gyneco) is a prospective multicenter cluster randomized controlled trial, with a stepped wedge design, to compare the perioperative management and outcomes before (usual care) and after the implementation of the ERAS protocol. We hypothesize that the adoption of the protocol will result in a reduction of the length of stay, complications, healthcare costs and will improve functional recovery and patient satisfaction.

Clusters are represented by all the regional hospitals with a gynecologic unit. Each cluster will progressively adopt the ERAS protocol according to a random order. At the end of the study, each unit will have a period of activity with usual procedures ("control period") and one with ERAS protocol adoption ("experimental period") with a cross-over like design, but with a single transition (from control to experimental).

The SPIRIT 2013 Checklist (https://www.spirit-statement.org/) is provided in the S1 File. Fig 1 shows the study diagram according to the CONSORT extension for Stepped Wedge Cluster Randomised Trials [18].

| Randomization | Study period (months) | | | | | |
| --- | --- | --- | --- | --- | --- | --- |
| group | 0-3 Baseline | 4-6 | 7-9 | 10-12 | 13-15 | Total |
| Group 1 | 120 | 120 | 120 | 120 | 120 | 600 |
| Group 2 | 120 | 120 | 120 | 120 | 120 | 600 |
| Group 3 | 120 | 120 | 120 | 120 | 120 | 600 |
| Group 4 | 120 | 120 | 120 | 120 | 120 | 600 |
| Number of patients by period: | | | | | | |
| • Control patients | 480 | 360 | 240 | 120 | 0 | 1200 |
| • Experimental patients | 0 | 120 | 240 | 360 | 480 | 1200 |

▭ Experimental period

Note: Due to COVID-19 outbreak the third period has been extended for three months (lasting 6 rather than 3 months) and the last study period shifted 3-months forward.

**Fig 1. Diagram of the ERAS-Gyneco Piemonte study with the expected number of patients in each group of hospitals and study period.**

The manuscript has been prepared according to the RECOvER checklist [14] (see the S1 File).

## Study organization

The study is promoted by the "S.Croce e Carle" Hospital (Cuneo) (previous promoter, until 19/02/2020, was the "Regina Montis Regalis" Hospital—Mondovì, Cuneo). The trial design, data collection and monitoring, statistical analyses and feedback activities are under the responsibility of the Clinical Epidemiology Unit of the "AOU Città della Salute e della Scienza di Torino"—Hospital (Torino) as part of the Work Package 3 of the EASY-NET Project.

## Inclusion criteria

- All gynecologic units of Piedmont Region hospitals performing at least 20 hysterectomies per year, with either laparoscopic or laparotomic approach.

- All patients hospitalized during the study period for elective hysterectomy for cervical or endometrial cancer or for benign uterine pathologies, with the exception of pelvic floor disorders.

## Exclusion criteria

- Units with less than 20 hysterectomies per year.

- Patients undergoing an emergency procedure.

- Patients with high complexity or clinical severity, to be documented at the time of admission, which represent contraindication to the application of the ERAS protocol (e.g. patients with ASA score V).

No exclusion will be allowed for patients with social issues, who do not allow for an adequate standard of personal autonomy or post-operative home care (conditions of severe dementia, physical dysfunction, state of indigence). The presence of these conditions will be recorded in the Case Report Form, but patients will be included and managed according to the ERAS protocol anyway. Specific attention will be required for the timing of discharge, which must be decided according to patient's needs and possibility (the date when the patient is "ready for discharge" will be recorded in the database).

### Stratification and randomization of the centers

All the gynecologic units were contacted prior to the beginning of the study to assess their level of knowledge of the ERAS protocol. Centers which already had fully adopted the protocol before the start of the study were excluded from randomization and included in an observational group to serve as an external benchmark for the study. All the other centers were stratified by the volume of hysterectomies performed during 2018 and randomly divided into 4 groups, each with a similar number of units and procedures. Then, the 4 groups were randomly ordered to activate the protocol according to a predefined calendar with 3 months interval between successive rounds. This stratified randomization was adopted to assure a more homogeneous composition of the groups for each activation period. All randomization procedures were performed by the Clinical Epidemiology unit after anonymizing the centers. The calendar date for protocol activation was communicated to each group about two months before the starting date to allow sufficient time for the training of the local ERAS team and to organize the activity.

After an initial 3-months period of data collection, performed in all centers to describe the baseline usual care, the first group of hospitals activates the ERAS protocol and then, every three months, the other groups follow step by step.

### Interventions

**ERAS group.** Each center is asked to appoint an "ERAS team", involving at least one person per professional role among gynecologic surgeons, anesthetists, nurses, and dieticians, with the aim to support the local implementation of the protocol and to be the point of reference for the center.

In the quarter preceding the date of randomization, each ERAS team receive specific training on the ERAS principles. The training consists of a one-day interactive course run by expert ERAS trainers. The selected participants are subsequently required to cascade training to their colleagues at local level. Slides are shared for this purpose and support by the expert ERAS trainers is available, if required.

**Control group.** Each center is assigned to the control group during the first three months of baseline and thereafter, until the randomization date. During the control period centers are required to continue with their usual perioperative care and to complete the Case Report Form (CRF) for the enrolled patients. Control group CRF will be analyzed to check compliance for each component of the ERAS protocol in the baseline phase and thereafter, until entering the experimental period.

**Audit&Feedback.** The study has a dedicated website (EPICLIN: https://new.epiclin.it/it/eras_isterectomia/) for data collection, data entry monitoring and feedback.

From the beginning of the study, all the centers can access the "monitoring area" on Epiclin to keep track of data collection progress and visualize graph and tables describing: number of enrolled and discharged patients, number of patients keen to participate to an interview after discharge and number of patients filling the quality of recovery questionnaire.

Once entering the experimental period, centers will also gain access to a feedback area. This area displays various graphs monitoring the indicators of compliance to all the ERAS items reported in S1 Table in S1 File. The aim is to verify the centers progress in applying the protocol in order to promptly identify low compliance issues and address them with corrective actions. Each indicator of adherence is stratified by study period (control and experimental) to appreciate changes in clinical practice. A radar graph allows to compare the adherence of groups of indicators measured in three different situations: before randomization, after randomization and in a benchmark setting (a group of regional hospitals already applying ERAS before the study).

After the first three months of ERAS adoption, the regional coordinating group organizes a workshop for each of the four groups with the expert ERAS trainers and the coordinating/data management team to discuss the feedback indicators, the critical issues encountered and to suggest possible concrete solutions to enhance compliance to ERAS protocol.

A periodic Newsletter is sent to all the ERAS teams to maintain engagement and motivation in the overall project and to share information on study progress and relevant news.

**Primary outcome.**   Mean LOS, calculated after excluding patients with LOS exceeding a predefined threshold, corresponding to the 98$^{th}$ percentile of the distribution.

**Secondary outcome.**

• Percentage of LOS longer than the threshold corresponding to the 98$^{th}$ percentile of the distribution

• Post-operative quality of recovery score, measured with the QoR-15 [16] questionnaire around 24 hours after surgery

• Incidence of post-surgical complications, defined according to the Clavien-Dindo classification [19]

• Intensive care unit access in the post-operative period

• Percentage of Emergency Department (ED) admissions up to 30 days after surgery (regardless of diagnosis)

• Percentage of readmissions to hospital within 30 days after surgery (regardless of diagnosis)

• Percentage of re-intervention (within 30 days after surgery)

• Perceived patient satisfaction score measured with the SSQ-8 [20] questionnaire via telephone interview around two weeks after discharge (only in a random sub-sample of patients keen to participate)

• Average healthcare costs, calculated from pre-hospitalization up to 30 days after surgery

• Assessment of professionals' satisfaction measured qualitatively.

**Data collection.**   The CRF is available in a dedicated area of the EPICLIN electronic platform. EPICLIN is developed and managed by the Clinical Epidemiology Unit, compliant with all the security requirements of the General Data Protection Regulation (EU regulation 2016/679). Data related to the peri-operative care are prospectively and uniformly collected in the database during the entire by the local ERAS team. The post-operative recovery will be

measured at 24 hours after the intervention through the QoR-15 questionnaire (available and validated in English) [16] and translated in Italian.

Patients' satisfaction is measured through the SSQ-8 questionnaire, available and validated in English [20] (and translated in Italian), administered by trained staff to a sample of patients (or alternatively their caregivers) during a telephone interview two weeks after discharge.

At the end of the study health professionals' satisfaction will be assessed qualitatively through questionnaires and focus groups.

Healthcare costs incurred between the first pre-admission visit and 30 days after hospital discharge, will be evaluated including the following categories of resources: pre-intervention visits, hospital stay days (including intensive care days), type of intervention, treatment of complications, re-interventions, ED access, new hospitalizations.

Further details on data collection are reported in the Supporting Information (see section "Data collection" in S1 File).

*Patient and Public Involvement*: No patient involved.

## Statistical plan

Fig 1 describes the sequence of randomizations of the clusters with the number of centers and the expected patients for the control and experimental periods. The total number of cases expected in 15 months is around 2400 patients (about 200 cases in the control period and 1200 in the experimental period).

Details on sample size estimation and power assessment are reported in the Supporting Information (see section "Statistical plan" in S1 File).

The main end-point, mean LOS calculated after excluding the durations greater than the threshold, will be compared between the two study periods using a random-effect linear regression model, adjusting for the time effect and the surgical technique (laparoscopic vs. open surgery). The center will be included in the model as a random effect. For dichotomous endpoints measured as proportions (e.g. length of stay above the threshold, complications, readmissions), the effect of implementing the ERAS protocol will be estimated with logistic regression models, with centers as random effects, including in the model the same set of covariates used for the analysis of the LOS.

Details on subgroup analyses and sensitivity analyses are provided in the Supporting Information (see section "Statistical plan" in S1 File).

## Ethics and dissemination

The study protocol has been approved by the Ethical Committee of the coordinating Center and by all participating centers. The study is conducted under the regulations of the Declaration of Helsinki.

All information collected during the trial will be kept strictly confidential. Information will be held securely at the Clinical Epidemiology Unit, accordingly to all aspects of GDPR 2018. The trial staff at the participating centers will be responsible for ensuring that any data or documentation sent to the Clinical Epidemiology Unit is appropriately anonymized. At the end of the trial, data will be securely archived for a minimum of 20 years.

Study results will be timely circulated within the hospital network and published in peer-reviewed journals, reported in line with the literature Consolidated Standards of Reporting Trials guidelines for stepped wedge cluster randomized trials [18] and the RECOvER checklist [14].

## Trial status

The trial is currently adhering to V.2.0 of the protocol (approved in January 2020). Enrollment was initiated on the 1st September 2019. Recruitment was initially expected to be completed in November 2020 but due to COVID outbreak is now expected to end in May 2021 (six months delay of the study timetable). At the beginning of May 2021 the number of patients enrolled is around 2400. Results will be available by the end of 2021.

## Discussion

This paper presents the study protocol of a stepped wedge cluster randomized controlled trial aimed to estimate the impact of a quality improvement intervention (the adoption of the ERAS protocol for hysterectomy in the entire regional hospital network in Piedmont). In 2017 an international survey [17] and a regional survey assessed that only few selected hospitals were compliant with the ERAS protocol in gynecologic surgery. Among the reasons for this limited uptake were the scarce diffusion of the multidisciplinary management of the patient and the lack of an organization that could enable the full adoption of the protocol, as a bundle, to obtain the desired positive effects.

To overcome the usual barriers to innovation in health care, especially at the organization level, and to monitor the changes in terms of appropriateness and safety, a structured Audit and Feedback (A&F) strategy has been planned in parallel to the trial. Even if the use of A&F is strongly recommended within the ERAS protocol, it is unusual that this strategy is carefully designed and conducted according to the best practice guidelines [21].

The "stepped wedge cluster randomized trial" study design has several advantages and some limits. Randomization at patient level would not be feasible due to the organizational nature of the intervention requiring a modification of the care process at center level. However, compared to other study designs, such as a before-after cluster randomized trial, the stepped wedge has a lower risk of bias due to possible confounding time effect and a complete coverage of the participating centers at the end of the study. This last advantage could theoretically represent even a risk in the unlikely, but not impossible, scenario of a disappointing effect of the experimental intervention.

Due to COVID-19 pandemic, the ERAS protocol was only partially implemented by some centers due to the shortage of staff and several organizational drawbacks. The extension of the study duration was intended to allow the centers, especially those in the third and fourth groups, to consolidate the implementation of the ERAS items. Such extension has also the purpose to get a sufficient sample size for the planned analyses on the secondary endpoints, counterbalancing the activity reduction (mainly among benign cases) registered during the pandemic.

To evaluate in a longer perspective both the effects and duration of the intervention and the impact of the COVID-19 pandemic, the time trend of some indicators (LOS, complications, readmissions) based on the routinely available data on hospital patient discharges will be analysed for the previous 5 years and monitored after study ending.

In conclusion, the main interest of the study results lies in the possibility to demonstrate positive clinical outcomes of the ERAS protocol even when its implementation is directed towards an entire regional network of gynecologic units, and not only towards selected and highly motivated ones. Moreover, the large cluster randomized trial reduces the risk of bias and the careful A&F strategy maximizes improvement of compliance. Due to these last aspects, the study will add useful evidence to the on-going EASY-NET project (http://easy-net.info/).

## Supporting information

**S1 File. ERAS protocol items for gynecologic surgery, synoptical scheme of the protocol, statistical plan, data collection, SPIRIT check list, RECOvER checklist.**
(DOCX)

**S2 File. ERAS hysterectomy protocol ENG 100522.**
(PDF)

**S3 File. ERAS hysterectomy protocol ITA 100522.**
(PDF)

## Acknowledgments

Thanks to Lisa Giacometti for English Language editing.

**ERAS-Gyneco Piemonte study group members** (*lead member: Eva Pagano, email: eva. pagano@cpo.it)

*Site investigators*: AO Alessandria: Vittorio Aguggia, Federica Borromeo; AO Maggiore della Carità, Novara: Katia Schipani, Daniela Surico; AO Mauriziano, Torino: Enrico Badellino, Annamaria Ferrero; AO S.Croce e Carle, Cuneo: Barbara Franzoso, Elisa Peano, Andrea Puppo; IRCCS Candiolo (TO): Riccardo Ponzone, Francesco Marocco; Nuovo Ospedale degli Infermi, Biella: Stefano Uccella, Chiara Violino; Ospedale Civile Agnelli, Pinerolo (TO): Andrea Bianciotto, Marco Canestrelli, Daniela Dompè; Ospedale degli Infermi, Rivoli (TO): Gabriele Molina; Ospedale Cardinal Massaia, Asti: Carlo Bocci, Paola Ferraris; Ospedale Castelli, Verbania: Riccardo Fiorentino; Ospedale Civico, Chivasso (TO): Maurizio Brusati; Ospedale Civile, Ciriè (TO): Antonio Alfeo, Mario Gallo, Romeo Geranio; Ospedale Civile, Ivrea: Fabrizio Bogliatto; Ospedale Maggiore, Chieri (TO): Massimo Mosetti, Giacomo Vaudano; Ospedale Martini, Torino: Marco Camanni, Elena Del Piano; Ospedale Regina Montis Regalis, Mondovì (CN): Alice Peroglio Carus, Elisa Piovano, Marta Sciandra; Ospedale S.Croce, Moncalieri (TO): Pier Luigi Montironi, Andrea Scoletta; Ospedale S.Giacomo, Novi Ligure (AL): Federico Tuo; Ospedale S.Lazzaro, Alba (CN): Enrica Bar, Alessandro Antonio Buda; Ospedale S.Spirito, Casale M.to (AL): Stefania Cigna Zorzetti, Francesco Lemut; Ospedale SS.Annunziata, Savigliano (CN): Alessio Garetto, Monica Mascher; Ospedale SS. Pietro e Paolo, Borgosesia (VC): Enrico Negrone, Elvira Sorbilli; Ospedale SS.Trinità, Borgomanero (NO): Roberto Biggiogera; AOU Città della Salute e della Scienza, Torino: Donato Mastrantuono, Claudio Plazzotta, Paolo Zola.

*Study coordination*: AO S.Croce e Carle, Cuneo: Andrea Puppo; Ospedale Martini, Torino: Marco Camanni, Elena Del Piano; AOU Città della Salute e della Scienza, Torino: Giovannino Ciccone, Eva Pagano*, Paolo Zola; Ospedale Regina Montis Regalis, Mondovì (CN): Elisa Piovano; AO Mauriziano, Torino: Paola Coata, Anna De Magistris, Barbara Mitola, Alessio Rizzo; Humanitas, Torino: Monica Rolfo, Laura Ceretto Giannone, Daria Bongiovanni; Ospedale S. Croce, Moncalieri (TO): Andrea Scoletta; Regione Piemonte: Anna Orlando; Rete Oncologica Piemonte e Valle d'Aosta: Oscar Bertetto.

*Technical staff*: AOU Città della Salute e della Scienza, Torino: Francesco Brunetti, Corinna Defilè, Vitor Hugo Martins, Lisa Giacometti, Matteo Papurello, Fabio Saccona, Danila Turco.

## Author Contributions

**Conceptualization:** Elisa Piovano, Eva Pagano, Elena Del Piano, Marco Camanni, Andrea Puppo, Giovannino Ciccone.

**Formal analysis:** Eva Pagano, Giovannino Ciccone.

**Methodology:** Eva Pagano, Giovannino Ciccone.

**Supervision:** Marco Camanni, Andrea Puppo, Giovannino Ciccone.

**Validation:** Elena Del Piano, Paola Coata, Daria Bongiovanni, Monica Rolfo, Laura Ceretto Giannone, Deliana Veliaj.

**Writing – original draft:** Elisa Piovano, Eva Pagano, Federica Rinaldi, Valentina Palazzo.

**Writing – review & editing:** Elisa Piovano, Eva Pagano.

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
