## [Decision Letter · Decision Letter 0]

6 Oct 2021

PONE-D-21-17975

Implementation of the ERAS (Enhanced Recovery After Surgery) protocol for hysterectomy in the Piedmont Region with an Audit&Feedback approach: study protocol for a stepped wedge cluster randomized controlled trial. A study of the EASY-NET project.

PLOS ONE

Dear Dr. Piovano,

Thank you for submitting your manuscript to PLOS ONE. After careful consideration, we feel that it has merit but does not fully meet PLOS ONE’s publication criteria as it currently stands. Therefore, we invite you to submit a revised version of the manuscript that addresses the points raised during the review process.

We look forward to receiving your revised manuscript.

Kind regards,

Luigi Maria Cavallo

Academic Editor

PLOS ONE

3. Thank you for stating the following in the Acknowledgments/ Funding Section of your manuscript:

“This work was supported by the Italian Ministry of Health and the Piedmont Region as part of the Easy-Net Project, grant number NET-2016-02364191.”

“The authors have declared that no competing interests exist.”

3. One of the noted authors is a group or consortium ERAS-Gyneco Piemonte study group. In addition to naming the author group, please list the individual authors and affiliations within this group in the acknowledgments section of your manuscript. Please also indicate clearly a lead author for this group along with a contact email address.

Additional Editor Comments (if provided):

The manuscript can be considered for publication after having competed revision as per reviewers requests.

Reviewers' comments:

Reviewer's Responses to Questions

**Comments to the Author**

1. Does the manuscript provide a valid rationale for the proposed study, with clearly identified and justified research questions?

Reviewer #1: Yes

Reviewer #2: Yes

2. Is the protocol technically sound and planned in a manner that will lead to a meaningful outcome and allow testing the stated hypotheses?

Reviewer #1: Yes

Reviewer #2: No

3. Is the methodology feasible and described in sufficient detail to allow the work to be replicable?

Reviewer #1: Yes

Reviewer #2: Yes

4. Have the authors described where all data underlying the findings will be made available when the study is complete?

Reviewer #1: Yes

Reviewer #2: Yes

5. Is the manuscript presented in an intelligible fashion and written in standard English?

Reviewer #1: Yes

Reviewer #2: Yes

6. Review Comments to the Author

You may also provide optional suggestions and comments to authors that they might find helpful in planning their study.

Reviewer #1: I have no comments to post. It looks well written and conceived. The analysis looks rigorous and satisfactory.

Reviewer #2: The manuscript is too long. We suggest to shorten it. Also revise English form

We suggest the authors to introduce a synoptical scheme of the protocol reported in the supplemental material to improve its understanding.

7. PLOS authors have the option to publish the peer review history of their article (what does this mean?). If published, this will include your full peer review and any attached files.

Reviewer #1: **Yes: **Felice Esposito

Reviewer #2: No

---

## [Author Response · Author response to Decision Letter 0]

14 Nov 2021

Please, find enclosed the rebuttal letter that responds to each point you raised.

---

## [Decision Letter · Decision Letter 1]

5 May 2022

Implementation of the ERAS (Enhanced Recovery After Surgery) protocol for hysterectomy in the Piedmont Region with an audit&feedback approach: study protocol for a stepped wedge cluster randomized controlled trial. A study of the EASY-NET project.

PONE-D-21-17975R1

Dear Dr. Piovano,

We’re pleased to inform you that your manuscript has been judged scientifically suitable for publication and will be formally accepted for publication once it meets all outstanding technical requirements.

Kind regards,

Luigi Maria Cavallo

Academic Editor

PLOS ONE

Additional Editor Comments (optional):

Reviewers' comments:

Reviewer's Responses to Questions

**Comments to the Author**

1. Does the manuscript provide a valid rationale for the proposed study, with clearly identified and justified research questions?

Reviewer #1: Yes

Reviewer #2: Yes

2. Is the protocol technically sound and planned in a manner that will lead to a meaningful outcome and allow testing the stated hypotheses?

Reviewer #1: Yes

Reviewer #2: Yes

3. Is the methodology feasible and described in sufficient detail to allow the work to be replicable?

Reviewer #1: Yes

Reviewer #2: Yes

4. Have the authors described where all data underlying the findings will be made available when the study is complete?

Reviewer #1: Yes

Reviewer #2: Yes

5. Is the manuscript presented in an intelligible fashion and written in standard English?

Reviewer #1: Yes

Reviewer #2: Yes

6. Review Comments to the Author

You may also provide optional suggestions and comments to authors that they might find helpful in planning their study.

Reviewer #1: The manuscript may be accepted for publication. I did not ask for any change at the time of the first review.

The Authors have submitted an improved version of the manuscript.

Reviewer #2: The authors completed reviewers' requests. the manuscript in this form is valid and address properly the topic.

7. PLOS authors have the option to publish the peer review history of their article (what does this mean?). If published, this will include your full peer review and any attached files.

Reviewer #1: **Yes: **Felice Esposito, MD, PhD, PhD

Reviewer #2: No

---

## [Editor Report · Acceptance letter]

18 May 2022

PONE-D-21-17975R1 

Implementation of the ERAS (Enhanced Recovery After Surgery) protocol for hysterectomy in the Piedmont Region with an audit&feedback approach: study protocol for a stepped wedge cluster randomized controlled trial. A study of the EASY-NET project. 

Dear Dr. Piovano:

I'm pleased to inform you that your manuscript has been deemed suitable for publication in PLOS ONE. Congratulations! Your manuscript is now with our production department. 

Kind regards, 

on behalf of

Dr. Luigi Maria Cavallo 

Academic Editor

PLOS ONE